# Recent Insights into the Functional Role of AMPA Receptors in the Oligodendrocyte Lineage Cells In Vivo

**DOI:** 10.3390/ijms24044138

**Published:** 2023-02-18

**Authors:** Maria Kukley

**Affiliations:** 1Achucarro Basque Centre for Neuroscience, 48940 Leioa, Spain; maria.kukley@achucarro.org; 2IKERBASQUE Basque Foundation for Science, 48013 Bilbao, Spain

**Keywords:** oligodendrocytes, NG2-glia, AMPA receptors, synaptic transmission, oligodendrocyte precursor cells, synapses, glutamate

## Abstract

This review discusses the experimental findings of several recent studies which investigated the functional role of AMPA receptors (AMPARs) in oligodendrocyte lineage cells in vivo, in mice and in zebrafish. These studies provided valuable information showing that oligodendroglial AMPARs may be involved in the modulation of proliferation, differentiation, and migration of oligodendroglial progenitors, as well as survival of myelinating oligodendrocytes during physiological conditions in vivo. They also suggested that targeting the subunit composition of AMPARs may be an important strategy for treating diseases. However, at the same time, the experimental findings taken together still do not provide a clear picture on the topic. Hence, new ideas and new experimental designs are required for understanding the functional role of AMPARs in the oligodendrocyte lineage cells in vivo. It is also necessary to consider more closely the temporal and spatial aspects of AMPAR-mediated signalling in the oligodendrocyte lineage cells. These two important aspects are routinely discussed by neuronal physiologists studying glutamatergic synaptic transmission, but are rarely debated and thought about by researchers studying glial cells.

## 1. Introduction

Receptors for α-amino-3-hydroxy-5-methyl-4-isoxazolepropionic acid (AMPARs) are ligand-gated ionotropic receptors for glutamate which are expressed by neurons and glia in the central and peripheral nervous system. As a matter of fact, when we think about AMPARs in neurons we tend to focus on their role in synaptic communication and plasticity, while when we think about AMPARs in glia, we tend to focus on their role as mediators of excitotoxic damage. Multiple studies especially emphasize the AMPAR-mediated damage to the oligodendrocyte (OL) lineage cells and show that blocking AMPARs reduces damage and increases survival of these glial cells during various pathological conditions and diseases. The functional role of AMPARs in the OL lineage cells during physiological conditions is discussed much less frequently, and often this topic simply remains in the shadow. One reason for this is that we still do not understand very well which processes are regulated or modulated by AMPARs in the OL lineage cells, and which molecular cascades and signalling pathways are triggered downstream of AMPAR activation in these glial cells during normal physiological conditions.

AMPARs are hetero-tetramers composed of four subunits (GluA1, GluA2, GluA3, and GluA4) that assemble together in various combinations to form functional receptors. The GluA1, GluA2, GluA3, and GluA4 proteins are encoded by the *Gria1, Gria2, Gria3,* and *Gria4* genes, respectively. The OL lineage encompasses cells at different stages of differentiation—from the OL progenitor cells (OPCs) to the myelinating OLs—both in the developing and in the adult brain (Figure 1). It is well established that OPCs (also known as glial cells expressing neural/glial antigen 2, or NG2^+^-glia) express functional AMPARs that mediate depolarizing inward current into the cell upon glutamate binding, and we summarized studies relevant to this topic in a recent review [1]. All four AMPAR subunits have been detected in OPCs, although their expression and properties seem to vary depending on the brain area, age of the animals, and the model system used to investigate the receptors [1]. Remarkably, multiple studies demonstrate that, similar to AMPAR-mediated synaptic communication between neurons, AMPAR-mediated synaptic signalling exists between neurons and OPCs in different areas of the rodent and human brain including the white matter [2,3,4,5,6,7,8,9,10,11,12,13,14,15,16,17,18,19,20]. The information regarding functional expression of AMPARs in the myelinating OLs remains less clear: some studies claim that AMPARs are present in these glial cells [21,22,23] while other studies fail to find them [4,24,25,26,27,28]. 

To understand the functional role of AMPARs in the OL lineage cells, several previous studies focused on testing how agonists and antagonists of AMPARs affect proliferation, differentiation, and morphology of these cells in vitro [29,30,31,32,33]. The major conclusion was that AMPARs are involved in the regulation of proliferation, migration, lineage progression, and morphological structure of the OL lineage cells. The discovery of neuron-OPC synapses in the hippocampus [2], and subsequently in many other regions of the rodent and human brain (as cited above), triggered an exciting idea that AMPARs in OPCs may acts as mediators of activity-dependent myelination. During the last two decades, new viral technologies have been developed, and new mouse lines have been generated that allow for targeting and manipulating proteins specifically in the OL lineage cells (for the list of available lines visit https://www.networkglia.eu/en/animal_models, accessed on 31 December 2022). Those tools prompted new research on the functional role of AMPARs in the OL lineage cells and the activity-dependent myelination in vivo, using deletion, overexpression, or modification of AMPAR subunits specifically in the OL lineage cells. This review discusses the most recent studies on the topic that have produced important and interesting discoveries. 

To collect the information for this review, searches were performed using the PubMed database and Google search. The following keywords were used in various combinations: oligodendrocyte, oligodendroglia, oligodendrocyte precursor cell, oligodendrocyte progenitor cell, OPC, NG2, myelin, myelination, glia, AMPA, glutamate, and kainate. During the search, no filters related to the publication date, country in which the study was performed, animal species, or experimental models were applied. However, this review aims to describe in detail only the results of studies carried out in vivo and which targeted AMPAR specifically in the OL lineage cells. All the studies found on the topic of the functional role of AMPARs in the oligodendrocyte lineage cells performed in vivo are included in the review. The studies which used local injection or systemic administration of AMPAR agonists or antagonists in vivo were not included because such experiments target AMPAR in all cells rather than focus specifically on the OL lineage cells. 

## 2. Loss-of-Function of AMPAR Subunits in OL Lineage Cells

### 2.1. In Vivo Experimental Systems

To date, two studies have investigated the functional role of AMPARs in the OL lineage cells in vivo using transgenic mice with deletion of one or several AMPAR subunits [15,34] (Table 1). 

Kougioumtzidou and colleagues used *Sox10*-Cre mice to inactivate (a) the *Gria2* gene only (Figure 2A,B), or (b) the *Gria2* gene on a *Gria3* germline knockout background, or (c) both *Gria2* and *Gria4* genes on a *Gria3* germline knockout background, specifically in the OL lineage cells [15]. The *Sox10* gene is SRY-box transcription factor 10 that belongs to a family of genes which play key roles in the formation of tissues and organs. In the CNS, Sox10 expression is restricted to the OL lineage cells, i.e., OPCs in the embryo and both OPCs and OLs in the postnatal mice [35]. The authors focused on the alterations in the subcortical white matter of the knockout animals. Evonuk and colleagues used PLP-Cre mice to inactivate the *Gria4* gene specifically in mature OLs in adult mice and investigated changes in the optic nerve as well as in the spinal cord white matter of normal mice and mice with experimental autoimmune encephalomyelitis (EAE) [34], (Table 1). The PLP, proteolipid protein, is a major myelin protein in the central and peripheral nervous system. It plays an important role in the formation or maintenance of myelin.

**Figure 2 ijms-24-04138-f002:**
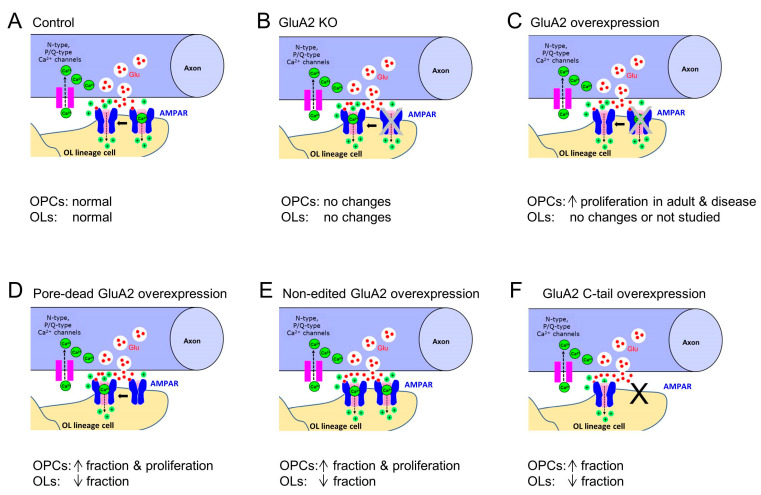
Brief summary of approaches used to study functional role of GluA2-containing AMPARs in the OL lineage cells and the corresponding experimental findings: (**A**) control; (**B**) knockout (KO) of GluA2 subunit in Sox10^+^-cells [15]; (**C**) overexpression of GluA2 subunit in OL lineage cells using different mouse lines [36]; (**D**) overexpression of pore-dead GluA2 subunit in proliferating callosal OPCs using retroviral gene delivery approach in vivo [13]; (**E**) overexpression of non-edited GluA2 subunit in proliferating callosal OPCs using retroviral gene delivery approach in vivo [13]; and (**F**) overexpression of GluA2 C-tail in proliferating callosal OPCs using retroviral gene delivery approach in vivo [13]. Red dots indicate glutamate molecules; green dots with the sign “+” indicate monovalent cations.

**Table 1 ijms-24-04138-t001:** Summary of the studies addressing the functional role of AMPAR subunits in the oligodendrocyte lineage cells in vivo.

Subunit	How AMPARs Were Targeted	In Which Cells AMPARs Were Targeted	Brain Areas Studied	How Targeting of AMPAR Was Verified	Consequences of AMPA Targeting	Reference
**Loss of function of AMPAR subunits in the OL lineage cells**						
GluA2	Conditional deletion of GluA2 by crossing Sox10-Cre and GluA2^flox/flox^ mice	OL lineage cells	Subcortical white matter	Electro-physiology: I-V curve of kainate-induced current changed from linear to inwardly rectifying	No change in the density of OPCs and OLs; no change in proliferation rate of OPCs	Kougioumtzidou et al., 2017 [15]
GluA4	Inducible deletion of GluA4 by crossing PLP-Cre and GluA4^flox/flox^ mice	OLs	Optic nerve and the spinal cord white matter	Immuno-labelling: reduced GluA4 expression in CC1^+^-OLs; Ca^2+^-imaging: reduction in glutamate-induced somatic Ca^2+^-responses	No changes in normal mice, but changes during EAE: better preservation of axons, reduced production of amyloid precursor protein, reduced microglia labelling	Evonuk et al., 2020 [34]
GluA4	Frameshift mutation in GluA4A using CRISPR/ Cas9 gene editing in zebrafish	OL lineage cells	Dorsal spinal cord	In situ hybridization: lower levels of GluA4A	Reduced net dorsal migration of OPCs; fewer myelin internodes per somite, and more myelin internodes per OL	Piller et al., 2021 [37]
GluA2, GluA3	Conditional deletion of GluA2 on a Gria3 germline KO background by crossing GluA3 KO, Sox10-Cre, GluA2^flox/flox^ mice	OL lineage cells	Subcortical white matter	Electro-physiology: reduced density of kainate-induced whole-cell current, I-V curve of kainate-induced whole-cell current changed from linear to inwardly rectifying; reduced number of synaptic contacts on OPCs	Reduced density of OLs at P7 and P14; reduced number of myelin figures at P14; increased apoptotic death of OL lineage cells at P7 and P14	Kougioumtzidou et al., 2017 [15]
GluA2, GluA3, GluA4	Conditional deletion of GluA2 and GluA4 on a Gria3 germline KO background by crossing Sox10-Cre, GluA2^flox/flox^, GluA4^flox/flox^ mice	OL lineage cells	Subcortical white matter	Electro-physiology: reduced density of kainate-induced whole-cell current; strongly reduced frequency of synaptic axon-glia EPSCs in OPCs	Reduced density of OLs at P14 and P53; increased apoptotic death of OL lineage cells at P14	Kougioumtzidou et al., 2017 [15]
**Gain-of-function and modification of AMPAR subunits in the OL cells**						
GluA2	Expression of the unedited GluA2(R583Q)-GFP subunit using retroviral gene delivery in vivo	Proliferating OPCs	Corpus callosum	Electro-physiology: I-V curve of axon-glia EPSCs changed from linear to inwardly rectifying; slight increase in quantal amplitude and single-channel conductance of AMPARs	Increased fraction of OPCs, decreased fraction of OLs, increased fraction of proliferating OPCs	Chen et al., 2018 [13]
GluA2	Expression of the pore-dead GluA2(R583E)-GFP subunit using retroviral gene delivery in vivo	Proliferating OPCs	Corpus callosum	Electro-physiology: I-V curve of axon-glia EPSCs changed from linear to inwardly rectifying; slight increase in quantal amplitude and single-channel conductance of AMPARs	Increased fraction of OPCs, decreased fraction of OLs, increased fraction of proliferating OPCs	Chen et al., 2018 [13]
GluA2	Expression of the cytoplasmic C-terminal (GluA2(813-862)) of GluA2	Proliferating OPCs	Corpus callosum	Electro-physiology: no changes	Increased fraction of OPCs, decreased fraction of OLs	Chen et al., 2018 [13]
GluA2	Inducible over-expression GluA2 subunit by crossing R26-Gria2 * and Sox10-CreER mice	OL lineage cells	Cortex, corpus callosum, external capsule	Calcium imaging in isolated brain OPCs: Ca^2+^ increases triggered in OPCs without GluA2 over-expression were much larger than in OPCs over-expressing the GluA2 subunit	No change in myelination in the young brain	Khawaja et al., 2021 [36]
GluA2	Inducible over-expression GluA2 subunit by crossing R26-Gria2 * and Cspg4-CreER	OPCs	Cortex, corpus callosum, external capsule	RT-qPCR: OPCs over-expressing GluA2 subunit express more Gria2 by ~3-fold	Increased proliferation of OPCs in adult mice; increased post-injury proliferation of OPCs and increased OLs lineage progression in a mouse model of neonatal hypoxia-ischemia and upon focal demyelination	Khawaja et al., 2021 [36]
GluA2	Inducible over-expression GluA2 subunit by crossing R26-Gria2 * and Plp1-CreER	OLs	Cortex, corpus callosum, external capsule		GluA2 overexpression in mature OLs does not prevent loss of OLs in a mouse model of neonatal hypoxia-ischemia	Khawaja et al., 2021 [36]

* R26-Gria2 mice is a new mouse line created by the authors. To create this line, the ROSA26 locus was targeted with a transgene containing CAG promoter, loxP-flanked STOP cassette, and EGFP-fused (Q/R) edited GluA2 (EGFP-Gria2) coding sequences.

Besides mice, zebrafishes have been used to study functional role of AMPARs in OL lineage cells. Piller and colleagues created transgenic zebrafishes with a global reduction in the GluA4a subunit and studied how this modification affects the behaviour of OPCs in the spinal cord [37] (Table 1). They used CRISPR/Cas9 gene editing to create a frameshift mutation resulting in a premature stop codon, which terminated the GluA4A polypeptide before translation of the critical glutamate-binding and transmembrane domains [37]. In their study, *Gria4a* was the subunit with unique expression in OL lineage cells but not in other cell types in the dorsal spinal cord of zebrafish at 72 hpf (hours post-fertilization). Hence, the role of the GluA4A subunit in the behaviour of the OL lineage cells could be investigated in these fishes. The authors focused on studying the migration of OPCs and myelination.

### 2.2. What Happens to Functional AMPARs in OL Lineage Cells upon Deletion of One or Multiple AMPAR Subunits? 

When animals lacking AMPAR subunits in OPCs (or other OL lineage cells) are generated, the first important task is to test how the expression and function of all AMPARs on the cell surface is altered in these animals. As OPCs receive AMPAR-mediated synaptic input from neurons, the second important task is to determine what happens specifically to those AMPARs that are located at the postsynaptic membrane of OPCs at axon–OPC synapses. Electrophysiology is one of the best approaches to address both questions because it allows direct examination of all functional AMPARs in a given cell, as well as of the synaptic AMPARs specifically. In addition, Ca^2+^-imaging is a valuable approach that allows testing for changes in Ca^2+^ permeability of AMPARs upon manipulation of AMPAR subunits in vivo. Immunohistochemistry for AMPAR subunits may also be used to test for changes in their expression in the OL lineage cells, but it is not always a straightforward approach because AMPAR labelling may appear distributed all over the slice if the subunit of interest is highly expressed at neuronal synapses; hence, it may be difficult to reliably distinguish neuronal vs. glial labelling. 

Kougioumtzidou and colleagues found that in the subcortical white matter, OPCs lacking the GluA2 subunit did not show alterations in the density of the whole-cell current triggered by kainate application (100 µM via a bath), but the current-voltage relationship was inwardly rectifying, whereas in the control mice it was linear [15] (Table 1). This pointed to an increased Ca^2+^ permeability of AMPARs, as expected [38]. In OPCs lacking the GluA2 and GluA3 subunits, the density of the kainate-induced whole-cell current was reduced to 53% compared with OPCs lacking only GluA3, and the current-voltage relationship was also inwardly rectifying [15] (Table 1). Notably, no up-regulation of the *Gria1* mRNA was detected in these mice using in situ hybridization, indicating that GluA1 was not upregulated as a compensatory mechanism. In OPCs lacking three AMPAR subunits (GluA2, GluA3, and GluA4), the kainate-induced whole-cell current was reduced to 23% compared with OPCs lacking two subunits (GluA2 and GluA3) [15] (Table 1). The current–voltage relationship of the kainate-induced current in these cells was linear, most likely because it was mediated by kainate receptors rather than AMPARs [15]. Taken together, these findings point to the fact that deletion of several AMPAR subunits in OPCs results in a reduced number of functional AMPARs expressed on the cell surface, whereas deletion of the GluA2 subunit results in altered Ca^2+^ permeability of AMPARs but probably not in their cell surface number. 

Deletion of the GluA4 subunit in mature OLs resulted in a substantially reduced GluA4 expression in the OLs labelled positively with anti-adenomatous polyposis coli clone CC1 antibody (CC1^+^-OLs), but not in astrocytes or neurons, within the spinal cord, as demonstrated by immunohistochemistry [34] (Table 1). Those OLs also showed reductions in the glutamate-induced somatic Ca^2+^-responses compared with OLs in control mice [34]. Introducing a frameshift mutation of the – *Gria4a* in zebrafish resulted in the lower levels of *Gria4a* demonstrated by in situ hybridization [37]. However, the authors also observed an enhanced expression of -*Gria4b* and *Gria2b*, indicating that there may be some compensation for the loss of a functional copy of the *Gria4a* gene [37].

To obtain information specifically about those AMPARs that are located at axon–OPC synapses, a possible approach is to record spontaneous (ideally miniature) axon–OPC synaptic currents and to analyse their amplitude and frequency. The amplitude of the currents provides qualitative information about the total number of AMPARs at synapses and their ion channel conductance. The frequency of the currents provides qualitative information about the release probability at axon–OPC synapses and the number of synapses made by neurons on the recorded cell. Analysis of these two parameters suggested that deletion of the GluA2 and GluA3 subunits in mouse OPCs does not alter the number and/or conductance of AMPARs at individual axon–OPCs synapses because the amplitude of spontaneous AMPAR-mediated axon–OPC synaptic currents did not change in these mice compared with animals lacking only the GluA3 subunit [15]. However, deletion of the GluA2 and GluA3 subunits resulted in a reduced number of synaptic contacts on OPCs, the conclusion being made based on the reduced frequency of axon–OPC synaptic currents and unchanged presynaptic release probability [15] (Table 1). Kougioumtzidou and colleagues also showed that in mice lacking three subunits of AMPARs (GluA2, GluA3, and GluA4) in OL lineage cells, the frequency of synaptic axon–glia EPSCs in OPCs was reduced more strongly than in mice lacking GluA2 and GluA3 subunits [15] (Table 1). The two studies targeting the GluA4 subunit in the OL lineage cells in mice [34] and zebrafishes [37] do not provide information about possible alterations of AMPARs at neuron–OPC synapses in these animals. Hence, it remains unclear whether and how their manipulations specifically affected AMPARs located at neuron–OPC synapses.

## 3. How Does Loss of Function of AMPARs Affect the Behaviour and Function of the OL Lineage Cells? 

### 3.1. Deletion of the GluA2 Subunit in Sox10^+^ OL Lineage Cells in Mice

Among AMPAR subunits, the GluA2 subunit is special because its properties determine the permeability of AMPARs for Ca^2+^-ions. The effect of the GluA2 subunit on the Ca^2+^ permeability of AMPARs depends on posttranscriptional editing at position 607 (Q/R site) in the GluA2 mRNA. If the edited GluA2 subunit incorporates into the AMPAR complex, it prevents Ca^2+^ influx. If the edited GluA2 is absent from the AMPAR complex, the receptor is Ca^2+^-permeable [38]. Hence, targeting the GluA2 subunit in the OL lineage cells is an interesting approach to find out how calcium influx through the GluA2-containing AMPARs and/or the downstream signalling pathways affect development and function of the OL lineage cells.

In the absence of only the GluA2 subunit in Sox10^+^ OL lineage cells in mice, development of the OL lineage cells appeared normal despite the altered Ca^2+^ permeability of AMPARs: the density of the OPCs labelled positively with platelet-derived growth factor receptor alpha antibody (PDGFR-alpha^+^ OPCs), the density of CC1^+^ OLs, and the proliferation rate of OPCs in the subcortical white matter did not differ in mice lacking the GluA2 subunit in OPCs vs. control mice with *Gria2*^+/+^ and/or *Gria2*^+/−^ OPC, when the animals were compared at P14 [15], (Figure 2A,B, Table 1). 

### 3.2. Reduction in GluA4 Subunit Expression in Zebrafish 

Piller and colleagues investigated how a frameshift mutation in the *Gria4* gene affects the OL lineage cells in zebrafish larvae [37]. Their time-lapse imaging experiments in vivo demonstrated that OPCs with *Gria4* mutation exhibited a significantly reduced net dorsal migration: OPCs migrated slower and for shorter distances (Table 1). As a result, fewer OPCs were found in the dorsal spinal cord in fishes with *Gria4* mutation vs. controls [37]. OPCs with the *Gria4* mutation also showed reduced migration in response to the local burst of glutamate trigged by uncaging of 4-methoxy-7-nitroindolinyl glutamate (MNI-glutamate), indicating that a reduction in GluA4A expression affects the ability of OPCs to sense glutamate and to migrate towards it [37]. Genetic mosaic experiments with transplantation of OPCs with the *Gria4a* mutation into the control fishes, and transplantation of control OPCs into the fishes with *Gria4a* mutation confirmed that the effects of GluA4A reduction on the migration of dorsal OPCs were cell-autonomous [37]. At the same time, the proliferation and survival of OPCs were unchanged in the fishes with *Gria4a* mutation compared with heterozygous and wild-type siblings [37]. 

Interestingly, the effects of *Gria4a* mutation on OPC migration persisted beyond the early developmental stages of the larvae: significantly fewer dorsal OL lineage cells (Sox10^+^) and more ventral OL lineage cells were still observed at 3 and 6 dpf (days post-fertilization) [37]. Mutation in the *Gria4a* gene also affected myelination in the developing spinal cord of the fishes: fewer myelin internodes per somite, and more myelin internodes per OL were found at 3–5 dpf in fishes with *Gria4a* mutation vs. control fishes using in vivo imaging (Table 1), although the average myelin internode length was not affected [37].

Migration of dorsal OPCs and the number of myelin internodes were improved by treatment of *Gria4a*-mutated larvae with the L-type voltage-gated Ca^2+^-channel agonist (±)-Bay K8644, suggesting that Ca^2+^ signalling induces OPC migration downstream of AMPAR activation [37]. This finding is in agreement with previous studies in mice which proposed that Ca^2+^ influx via voltage-gated Ca^2+^-channels downstream of AMPAR-mediated glutamate signalling is an important regulator of OPC development [32,39,40,41].

Taken together, these findings show that GluA4A-containing AMPARs in concert with voltage-gated Ca^2+^- channels are involved in regulation of OPC migration and myelination in the dorsal spinal cord of zebrafish larvae.

### 3.3. Deletion of the GluA4 Subunit in PLP^+^ Oligodendrocytes in Mice

Inducible deletion of only the GluA4 subunit of AMPARs in PLP-expressing OLs in adult (9-week-old) mice did not affect the total number of axons, the proportion of myelinated and unmyelinated axons, axonal diameters, or the g-ratio (ratio of axon diameter to axon and myelin diameter) in the lumbar ventral white matter of the spinal cord [34]. Hence, in naïve mice, these parameters seem not to depend on the function of the GluA4 subunit in mature OLs. However, AMPARs containing the GluA4 appear to play a negative role during demyelination as the clinical scale during EAE was better in mice lacking the GluA4 subunit in mature OLs compared with control animals [34]. The GluA4-lacking mice had a better preservation of axons during EAE: they showed a higher total number of axons, a higher number of myelinated axons estimated using toluidine blue staining and electron microscopy, as well as a higher optical density of myelin staining (Table 1). These mice also had reduced production of amyloid precursor protein, which is a marker of axonal injury, in the dorsal (although not ventral) spinal cord. The percentage of unmyelinated axons during EAE was lower in mice lacking GluA4 compared with control animals, suggesting that they had less demyelination. In addition, mice lacking the GluA4 in OLs demonstrated reduced microglia labelling in the lumbar spinal cord white matter during EAE, probably because the inflammation was less severe [34] (Table 1). 

Taken together, these experiments show that GluA4-containing AMPARs in mature OLs contribute to damage of axons and myelin during EAE, a mouse model of multiple sclerosis. Calcium entry through GluA4-containing AMPARs may mediate the damage. Alternatively, or additionally, other mechanisms of cell injury may be triggered upon activation of AMPAR in OLs, e.g., AMPAR-induced opening/potentiation of NMDA receptors in OLs or calcium-induced calcium release from the internal stores. 

### 3.4. Deletion of Multiple AMPAR Subunits in Sox10^+^ Cells in Mice

In the absence of both the GluA2 and the GluA3 subunits in Sox10^+^ cells in mice, a statistically significant reduction in the density of OLs by ~22% and ~27% was observed at P7 and P14, respectively, based on immunohistochemistry with a CC1 marker. At P21 -P70, a tendency for reduced density of OLs remained in mice lacking GluA2 and GluA3 vs. control animals, but the difference was not statistically significant [15]. At the same time, P3–P70 mice lacking GluA2 and GluA3 in OPCs did not demonstrate significant differences in the division rate or density of PDGFR-alpha^+^ OPCs in the subcortical white matter, suggesting that the lower density of OLs was not a result of a reduction in the division rate or differentiation of OPCs [15]. However, deletion of the GluA2 and GluA3 subunits in OPCs triggered increased apoptotic death of the OL lineage cells, based on the immunohistochemistry with anti-cleaved Caspase-3 and anti-Olig2 (oligodendrocyte transcription factor 2) in P14 mice [15]. In accordance with the lower density of OLs, there was a ~20% reduction in the number of myelin figures in P14 mice lacking the GluA2 and GluA3 subunits. However, the number and length of myelin internodes made by individual OLs, as well as the g-ratio, did not differ in these mice compared with the control animals (Table 1). At P70, the differences in the number of myelin figures were no longer observed, most likely because the alterations in the density of CC1^+^ OLs diminished at that age [15]. 

In the absence of the three AMPAR subunits (GluA2, GluA3, and GluA4), there was a ~22% and ~26% reduction in the density of CC1^+^ OLs at P14 and P53 (Table 1), respectively, but the density of OPCs was unaltered compared with mice lacking GluA3 in all cells [15]. Similar to mice lacking only two subunits of AMPARs in OL lineage cells, mice lacking three subunits also showed a higher proportion of Olig2^+^ cells that expressed cleaved Caspase-3. At the same time, there were no significant differences in the average number or length of myelin internodes in individual OLs, as demonstrated by filling single OLs with a dye in brain slices [15].

Together, these data suggest that AMPAR-mediated signalling in the OL lineage cells supports survival of newly-differentiating OLs during the first two months of mouse life but does not affect the properties of myelin sheaths made by individual OLs. Furthermore, the survival of OL lineage cells may depend on the subunit composition of their AMPARs. It seems necessary that the receptors are assembled as heterodimers of at least two different subunits, rather than as homodimers. 

## 4. Gain-of-Function and Modification of AMPAR Subunits in OL Cells

### 4.1. In Vivo Experimental Systems

To date, two studies have investigated the functional role of AMPARs in the OL lineage cells in vivo using gain-of-function and modification of the AMPAR subunit GluA2 [13,36] (Figure 2C–F, Table 1). In our study [13], we targeted the GluA2 subunit in the proliferating OPCs located in the corpus callosum during the peak of callosal myelination in mice (second-third postnatal week). At that point of mouse development, many callosal OPCs contain the edited GluA2 subunit, and their AMPARs are Ca^2+^-impermeable [11,13,14]. Using a retroviral gene delivery approach, we expressed (a) the unedited GluA2 subunit, or (b) the pore-dead GluA2 subunit of AMPARs, or (c) the cytoplasmic C-terminal (GluA2(813–862)) of the GluA2 subunit (C-tail) in proliferating callosal OPCs (Figure 2D–F). The first modification aimed to increase the Ca^2+^ permeability of AMPARs. The second modification aimed to reduce the current flow through the AMPARs. The third modification was designed to affect the interaction between the GluA2 subunit and the AMPAR-binding proteins and to perturb the trafficking of the GluA2-containing AMPARs [13]. We focused on analysing the changes in proliferation and differentiation of OPCs in the mouse corpus callosum during the second and third postnatal week upon these manipulations of AMPARs.

Recently, another study used a different approach to target the GluA2 subunit in OPCs in vivo [36] (Figure 2C). They created new mouse lines with an inducible cell-specific overexpression of the edited GluA2 subunit in Sox10^+^-cells, or in cells expressing chondroitin sulfate proteoglycan 4 (Cspg4^+^-cells), or in cells expressing proteolipid protein 1 (PLP1^+^-cells), aiming to target all cells of the OL lineage, or specifically OPCs, or specifically OLs, respectively [36]. The authors focused on studying the effects of edited GluA2 overexpression in the OL lineage cells in the cortex, corpus callosum, and external capsule of juvenile and adult normal mice, as well as in mice with hypoxic-ischemic brain injury or demyelination.

### 4.2. What Happens to Functional AMPARs in OL Lineage Cells upon Overexpression and Modification of AMPAR Subunits? 

To test how the function of AMPARs is altered in OPCs with virally expressed non-edited or pore-dead GluA2 subunits, or the GluA2 C-tail, we used electrophysiology [13]. In OPCs expressing the non-edited GluA2 subunit, the amplitude of quantal synaptic axon–glia currents was slightly larger than in the control OPC (i.e., those with virally expressed green-fluorescent protein only), and the current–voltage relationship of the evoked axon–glia currents was inwardly rectifying, indicating that AMPARs were Ca^2+^-permeable (Table 1). These properties are expected for AMPARs lacking the edited GluA2 subunit [38]. In OPCs expressing the pore-dead GluA2 subunit, we anticipated recording axon–glia currents with smaller amplitudes in analogy to previous observations in cultured pyramidal neurons [42]. However, it was not the case, and the changes in synaptic AMPARs were comparable to those occurring with non-edited GluA2. We observed quantal synaptic axon–glia currents with larger amplitudes and an inwardly rectifying current–voltage relationship [13]. A possible explanation of this phenomenon is that the expression of pore-dead GluA2 mimicked the absence of functional GluA2 from the AMPARs complex. Accordingly, we observed larger single-channel conductance of AMPARs containing non-edited or pore-dead GluA2 subunit, as described previously in model systems [43], which is in line with the inwardly rectifying current–voltage relationship, and likely underlies the larger current amplitude. Expression of the GluA2 C-tail did not affect the amplitude of quantal axon–glia currents, or the single-channel conductance of AMPARs, or the current–voltage relationship [13], (Table 1). This suggests that expression of the GluA2 C-tail does not alter the ionotropic function of AMPARs at axon–OPC synapses. In our experiments, we did not study the total surface expression of AMPARs in OPCs. Hence, we do not know whether all the observed effects on the development of the OL lineage cells (described below) are mediated only by altered AMPARs at axon–OPC synapses, or also by AMPARs located in non-synaptic cellular compartments, e.g., the cell soma. 

Khawaja and colleagues used a different approach to examine alterations of AMPAR function in OL lineage cells upon GluA2 overexpression (Table 1). They isolated OPCs from P3-P10 control mice and mice with GluA2 overexpression in OL lineage cells, and performed Ca^2+^ imaging in a dish. Upon co-application of AMPA and cyclothiazide (an inhibitor of AMPAR desensitization), calcium increases triggered in OPCs overexpressing the GluA2 subunit were much smaller than in OPCs without GluA2 overexpression [36], indicating that GluA2 overexpression reduced the permeability of AMPARs for Ca^2+^ ions as expected [38]. The described experimental design has a certain disadvantage, though: the OPCs were isolated from the whole brain without dissecting specific brain regions, whereas it is known that the Ca^2+^ permeability of AMPARs in OPCs in normal animals differs between the brain regions. For example, in juvenile animals, AMPARs in hippocampal OPCs are Ca^2+^ permeable [2], whereas in callosal OPCs they are Ca^2+^ impermeable [11,13,14]. In the experiments of Khawaja and colleagues, it is not known whether OPCs recorded during Ca^2+^ imaging expressed or lacked the edited GluA2 subunit, and it is also not known whether the overexpression of GluA2 occurred in OPCs originally expressing or lacking the edited GluA2 subunit. Additional experiments studying the properties of AMPARs upon overexpression of the GluA2 subunit would be useful and interesting, e.g., single-cell electrophysiology or Ca^2+^ imaging in defined regions in brain slices.

### 4.3. How Does the Overexpression and Modification of AMPAR Subunits Affect the Behaviour and Function of OPCs? 

#### 4.3.1. Targeting the GluA2 Subunit of AMPARs in Proliferating OPCs 

In the corpus callosum of mice expressing a non-edited or pore-dead GluA2 subunit, or the C-tail of the GluA2 subunit in dividing OPCs, the proportion of CC1^+^ OLs (as well as the proportion of OLs labelled positively for the myelin-associated glycoprotein (MAG^+^ OLs)) among all virally targeted cells was lower than in control mice, whereas the proportion of NG2^+^ OPCs was higher [13] (Figure 2D–F). Expression of the non-edited or pore-dead GluA2 subunit, but not the GluA2 C-tail, in callosal OPCs increased their proliferation and prompted the OPCs to re-enter mitosis [13] (Figure 2D,E, Table 1). Neither of the modifications specifically triggered the subsequent differentiation of those OPCs which re-entered mitosis. Hence, properties of the GluA2 subunit in dividing OPCs appear to be important for supporting the balance between OPCs and OLs by modulating the “decision” of OPCs to stay in or re-enter mitosis, vs. the “decision” to differentiate. As expression of the non-edited or pore-dead GluA2 subunit alters the Ca^2+^ permeability of AMPARs, it is tempting to suggest that downstream signalling pathways mediating the effects of AMPARs may involve Ca^2+^-dependent mechanisms. Those mechanisms may not necessarily be limited to Ca^2+^ entry through AMPARs themselves, but may involve AMPAR-triggered opening of the voltage-gated Ca^2+^ channels or Ca^2+^-induced Ca^2+^ release from the internal stores, etc. Interestingly, when the C-tail of the GluA2 subunit was expressed in OPCs, no alterations in the ionotropic function of AMPARs were observed, but the balance between OPCs and OLs was changed, whereas the predisposition of OPCs to re-enter mitosis was not. This suggests that non-ionotropic function of AMPARs may be part of some mechanisms regulating proliferation and differentiation of OPCs, but not the others. 

#### 4.3.2. Targeting the GluA2 Subunit of AMPARs in OL Lineage Cells 

In mice overexpressing the GluA2 subunit of AMPARs in a larger population of the Cspg4^+^-OPCs (rather than only in the proliferating ones, as in our study), the percentage of CC1^+^ OLs and PDGFR-alpha^+^ OPCs, as well as EdU^+^ (5-ethynyl-2′-deoxyuridine) OPCs in the corpus callosum, cortex, or external capsule did not differ from the control mice at early developmental ages, i.e., before P24 [36]. In mice overexpressing the GluA2 subunit of AMPARs in Sox10^+^-cells (encompassing OPCs and OLs), the levels of myelin proteins, g-ratio, and fraction of myelinated axons did not differ from the control mice at early developmental ages [36]. A possible explanation of the lack of effects may be the fact that expression of the edited GluA2 subunit in OPCs is already high during the first three postnatal weeks, at least in the corpus callosum [11,13,14], and overexpression of the edited GluA2 does not bring any detectable changes. In line with the latter idea, in adult mice, when the levels of the edited GluA2 subunit in OPCs are low [14], the Cspg4^+^-specific GluA2 overexpression increased proliferation of OPCs [36] (Figure 2C, Table 1), but the proportion of CC1^+^ OLs was not altered upon GluA2 overexpression. 

The properties of AMPARs in OL lineage cells, specifically their Ca^2+^ permeability, are expected to play an important role during pathologies, but the mechanisms may be cell-stage specific. For instance, Khawaja and colleagues found that in a mouse model of neonatal hypoxia-ischemia, Cspg4^+^-specific GluA2 overexpression promoted post-injury proliferation of OPCs and stimulated OL lineage progression which was reduced by injury [36] (Table 1). However, the PLP1^+^-specific GluA2 overexpression did not prevent the loss of CC1^+^ Olig2^+^ OLs in the neonatal hypoxia-ischemia model [36] (Table 1). The Cspg4^+^-specific GluA2 overexpression in adult mice resulted in enhanced proliferation OPCs and OL cell lineage progression upon focal demyelination injury in the corpus callosum (Table 1). Taken together, these findings suggest that targeting the GluA2 subunit in OPCs but not in mature OLs may be beneficial for protection from AMPAR-mediated excitotoxicity during neonatal hypoxia-ischemia injury as well as during demyelination in the adult brain.

## 5. Agreements and Contradictions Regarding the Functional Role of AMPARs in OL Lineage Cells

The studies addressing the functional role of AMPARs in OL lineage cells have provided valuable information but, taken together, the experimental findings still do not provide a clear picture on the topic.

Manipulations of the GluA2 subunit of AMPARs in the OL lineage cells in vivo (with the exception of the GluA2 C-tail expression) triggered alterations in Ca^2+^ permeability of AMPARs and resulted in changes of proliferation and/or differentiation of OPCs in some models [13,36], but not in others [15,36]. The significance of the GluA2-dependent Ca^2+^ permeability of AMPARs for proliferation and differentiation of OPCs remains unclear because various studies showed that higher Ca^2+^ permeability may cause no changes [15], may result in increased proliferation and decreased differentiation of OPCs [13], or may result in low proliferation and un-altered differentiation of OPCs [36]. Manipulations targeting the GluA4 subunit of AMPARs in mice also altered Ca^2+^ permeability, but the results suggested that in the mouse spinal cord, the GluA4-containing AMPARs are without significance for mature OLs during physiological conditions but play a role during demyelination [34]. In zebrafish larvae, however, the GluA4-containing AMPARs are involved in regulation of OPC migration and myelination in the spinal cord [37]. Manipulations targeting multiple subunits of AMPARs altered Ca^2+^ permeability and affected survival of newly differentiating OLs, but did not have any effect on the properties of myelin sheaths made by individual OLs [15].

If an experimental manipulation of AMPARs triggers alterations in Ca^2+^ permeability, it is tempting to suggest that this is a mechanism underlying the observed changes in the OL lineage cells. An important issue to consider in this regard is that even if the Ca^2+^ permeability of AMPARs is reduced, or AMPARs themselves are Ca^2+^ impermeable, their activation may still trigger Ca^2+^ entry into the cell because AMPAR-dependent membrane depolarization may open voltage-gated Ca^2+^ channels [44]. On the other hand, voltage-gated and leak potassium channels expressed in OPCs may counteract AMPAR-dependent membrane depolarization [44] and Ca^2+^ entry. Furthermore, other proteins which act in concert with AMPARs, e.g., auxiliary transmembrane AMPAR related proteins (TARPs) [6] and various molecules downstream of AMPAR activation, also likely contribute to the AMPAR-dependent modulation of behaviour and function of the OL lineage cells. Hence, upon targeting AMPARs experimentally, it would make sense to evaluate not only the functional alterations and Ca^2+^ permeability of AMPARs themselves but also possible changes in the expression and function of Ca^2+^ and K^+^ channels and other “AMPAR partner proteins” which may compensate for or enhance the modulation of cellular events triggered by AMPAR deletion or overexpression. It would be also useful to learn about the temporal dynamics and spatial distribution of AMPAR partner proteins, e.g., the dependence of their expression and function on animal age, brain region, etc. during normal conditions, because these factors may be important for interpretation of the experimental findings obtained upon manipulations of AMPAR subunits in vivo. 

## 6. Temporal Aspects of AMPAR-Mediated Signalling in OL Lineage Cells

Among all glial cells in the healthy CNS, only OPCs (see references above) and cerebellar Bergmann glia [45] receive direct AMPAR-mediated synaptic input from neurons. An intriguing question is what the specific role of those AMPARs which are located at neuron-glia synapses is. Glutamate is the major excitatory neurotransmitter in the CNS, which is released from axons during neuronal activity and binds to AMPARs. Therefore, many studies put forward the idea that AMPARs at neuron–OPC synapses are among the mediators of activity-dependent myelination. Neuron–OPC synapses may be the important contact points between the two cell types which give neurons a possibility to regulate proliferation and differentiation of OPCs, as well as myelination, in an activity-dependent fashion. There is, indeed, clear evidence that neuronal activity modulates behaviour of OPCs and myelination [46,47]. Furthermore, neuronal modulation of OPCs’ behaviour is not an all-or-none signal, but may depend on the firing pattern of neurons, i.e., some patterns of neuronal activity are more likely to promote proliferation of OPCs while others are more likely to promote their differentiation [12]. Yet, so far, there has been no direct demonstration that specifically the synaptic AMPARs mediate this modulation. This important question is certainly of great value, and would be interesting to address in future studies. It is worth noting that the functional role of axon–glia synapses is likely not restricted to their (possible) role in myelination because cerebellar Bergmann glia receive synaptic input from neurons as well but do not belong to the myelinating cell lineage.

Remarkably, neuron–glia synaptic signalling mediated by AMPARs is rapid (on a timescale of milliseconds, similar to neuronal synapses (Figure 3A,B)), brief, and precisely directed only to those glial cells with which a neuron forms synapses. However, so far, studies addressing the functional role of AMPARs in OPCs (including ours) have focused only on investigating the slow cellular events (proliferation, differentiation, and myelination), which occur on a timescale of minutes and probably days. A puzzle which remains is why neuronal axons invest energy into building and keeping complicated presynaptic neurotransmitter release machinery for fast signalling with glial cells if the goal is to modulate only the slow cellular events in glia. One possibility is that it is not the fast neuron–glia synaptic signalling per se that is important for the modulation of slow events in glia, but the short-term [12] and/or long-term [3] plasticity of neuron–glia synapses that are both slower processes (i.e., may occur on a time-scale of seconds, minutes, and probably hours). Another possibility (which does not exclude the first one, though) is that rapid neuron–glia synaptic signalling is important for the modulation of some fast events in individual glial cells or in glial ensembles which we still have not been thinking about, and perhaps have not even discovered yet. In neurons, the speed of synaptic transmission determines the communications rate between the cells, and influences local circuit dynamics [48]. However, the functions and the dynamics of neuron–glia synaptic circles remain understudied, and a strong necessity exists to design experiments addressing these questions. This also brings up the thinking about possible functions of OPCs beyond their role as oligodendroglial progenitors, and about possible functional similarities between OPCs (which are within the OL lineage cells) and cerebellar Bergmann glia (which do not belong to the myelinating cell lineage). Why only these two types of glia receive fast synaptic input from neurons in the healthy CNS remains an open question. Recent studies showing that AMPAR-mediated synaptic input also exists between neurons and glioma cells [49,50] raise further questions about why glioma cells also receive synaptic input from neurons and what similarities there are between glioma cells, OPCs, and Bergmann glia.

Although it is well established that AMPAR-mediated signalling between neurons and OPCs is synaptic and fast (on a time-scale of milliseconds) (Figure 3B), the properties of AMPAR-mediated signalling between neurons and mature OLs have only started to be revealed, and it is still even disputed in which brain region the mature OLs express AMPARs and at which membrane compartments their AMPARs may be located. We and others did not detect AMPAR-mediated synaptic currents in mature OLs using patch-clamp recordings [4,28]. This may suggest that (a) mature OLs do not express AMPARs, or (b) AMPAR-mediated signalling between axons and mature OLs is non-synaptic, or (c) voltage- and/or space-clamp errors occurring during patch-clamp recordings [51] prevented detection of the AMPARs and correct determination of the mode of AMPAR-mediated axon–OL signalling. The latter may especially be true if AMPARs are located within the myelin sheathes, i.e., far from the OL cell soma where the recording patch pipette is placed. Hence, at present it is not possible to make a conclusive statement regarding the mode and the timing of AMPAR-mediated signalling between axons and mature OLs. 

Some years ago, an interesting concept of axo-myelinic synapse and axo-myelinic neurotransmission was introduced [23,52]. This concept suggests that as an action potential travels along the axon, it triggers glutamate release from the axon under the myelin, and the released glutamate activates AMPARs and/or NMDA receptors on the myelin membrane followed by Ca^2+^ entry into the cytosolic compartment of myelin [52]. In this way, mature OLs and myelin may detect and respond to neuronal activity. Of course, it is interesting and important to understand whether axo-myelinic synapses, neuron–OPC synapses, and classical neuron–neuron synapses all have the same properties. For neuronal physiologists, a key parameter at classical neuronal synapses is timing. Chemical synaptic transmission between neurons is rapid and brief, i.e., it occurs on a time-scale of a few milliseconds (Figure 3A). The timing of synaptic transmission is important because it determines the speed of communication between neurons and the speed of information processing within neuronal circuits. With regard to timing, neuron–OPC synapses are identical to classical neuronal synapses because release of glutamate from neurons, its binding to synaptic AMPARs in OPCs, and subsequent AMPAR-mediated synaptic current in OPCs all occur with a few milliseconds at maximum (Figure 3B). The timing of neurotransmission at axo-myelinic synapses seems to be slower and to have different properties [23]. At this synapse, glutamate is released from the internodal part of the axons in a vesicular manner, but the Ca^2+^ source for vesicular fusion is provided by ryanodine receptors on axonal Ca^2+^ stores controlled by L-type voltage-gated Ca^2+^ channels activated by the depolarization of the internodal axolemma [23]. The likely reasons underlying the slower timing of axo-myelinic synapses vs. classical neuronal and neuron–OPC synapses are: (a) more intermediate steps are required between the action potential invasion and neurotransmitter release (i.e., voltage-gated L-type Ca^2+^-channels are slower, and store-operated channels have to switch on as well); and (b) the location of the AMPARs. If the AMPARs directly face the periaxonal space, the axon-myelinic signalling may not be very slow, but if the AMPARs are located within the myelin, the glutamate released from axons would have to diffuse a much longer distance to reach the AMPARs and NMDA receptors within the myelin, and the signalling would be slow. Taken together, the present state of our knowledge leads us to conclude that axo-myelinic synapses operate on a much slower time-scale than classical neuron–neuron or neuron–OPC synapses, and the mode of transmission at axo-myelinic synapses is not fully clear (Figure 3C). 

## 7. Spatial Aspects of AMPAR-Mediated Signalling in OL Lineage Cells

Besides the temporal aspects of AMPAR-mediated signalling in the OL lineage cells, other important issues include the spatial stability of AMPARs in OL cells, their trafficking to and from the postsynaptic sites of neuron–glia synapses, as well as the stability of glutamatergic neuron–glia synapses themselves.

In vivo imaging showed that many OPCs undergo multiple cell divisions, and many OPCs also migrate, translocating their cell bodies from day to day in three dimensions [53]. This raises the question of whether AMPAR-containing neuron–OPC synaptic junctions are stable, and what happens to them when OPCs migrate or divide. We know that OPCs have synapses with neurons even during mitosis [18,20], but we do not know whether these synapses have been established once and remain stable for a long-time, or whether neuron–OPC synapses may assemble and disassemble on a relatively fast time-scale. It also remains unclear whether the disassembled synapses would reassemble at exactly the same previous locations on the OPC membrane or at different ones. This is an important question because AMPAR-mediated functions in OPCs likely depend on the integration of neuronal input by OPCs [44,54], while the result of the integration may depend on the location of individual synapses on the OPC membrane and their distance from the cell soma. 

In neurons, AMPARs are dynamic, meaning that they are constantly trafficking from and into the postsynaptic membrane [55]. Information about AMPAR trafficking in OPCs, neuron–OPC synapses, and myelin is currently not available. There is a great necessity to design experiments that address this question because AMPAR-dependent modulation of proliferation, differentiation, and migration of OPCs, survival of OLs, and perhaps other cellular events may depend on the diffusion and/or mobility of AMPARs. Studying the trafficking of AMPARs in OPCs, at neuron–OPC synapses, and in the myelin is also important for understanding the role of AMPARs in activity-dependent myelination. We have previously shown that different patterns of neuronal activity affect OL lineage cells in a distinct way [12]. However, the effects of neuronal activity on the OL lineage cells may be more complex because it is possible that even the same patterns of neuronal activity influence OPCs and/or OLs in distinct ways, depending on the subunit composition, trafficking and persistence of surface AMPARs at neuron–glia synapses and/or in the myelin. We have recently reviewed several state-of-the-art approaches which have been used for studying AMPARs in neurons and that can be applied to further investigate the properties and functions of AMPARs in OL lineage cells [1]. It will be especially interesting to study the oligodendroglial AMPARs in freely moving animals and also in combination with behavioural paradigms.

## Figures and Tables

**Figure 1 ijms-24-04138-f001:**
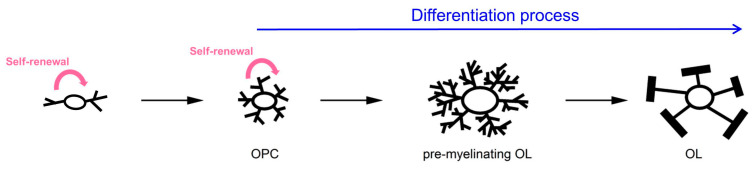
Scheme of the oligodendrocyte lineage cells. Note that early and later OPCs are capable of self-renewal, while more mature OL lineage cells are not.

**Figure 3 ijms-24-04138-f003:**
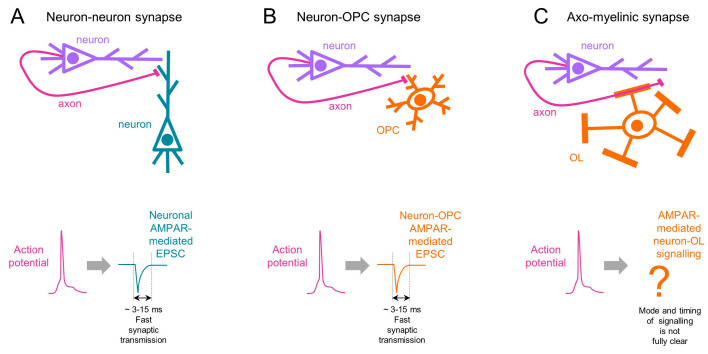
Timing of neurotransmission at (**A**) neuronal synapses, (**B**) neuron–OPC synapses, and (**C**) axo-myelinic synapses. The upper panels are a schematic drawing of the connections between (**A**) two neurons, (**B**) a neuron and an OPC, and (**C**) an axon and an OL. The corresponding lower panels show the action potential in the axons (spike in magenta colour), and the corresponding fast excitatory postsynaptic current (EPSC) at neuronal and neuron–OPC synapses (blue color). The mode of signalling at axo-myelinic synapses is not fully clear, but appears to be slower than at classical synapses.

## Data Availability

Data sharing is not applicable to this article as no new data were created or analyzed in this study.

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
