# Peer review of "Recent Insights into the Functional Role of AMPA Receptors in the Oligodendrocyte Lineage Cells In Vivo"

_ijms, 2023, doi:10.3390/ijms24044138_

Round 1

Reviewer 1 Report

This review article written by Kukley mentions the comments, problems and prospects of recent research about the functional role of AMPA receptors in oligodendrocyte (OL) lineage cells (Fig. 1) in vivo by mainly focusing on GluA2 (Fig. 2) and GluA4.  Moreover, synapse between OL lineage cell and neuron is compared in timing with neuronal synapse (Fig. 3).  This manuscript is well written and seems to be interesting.  There are several points that should be addressed and may serve to amend this manuscript, as follows:

Major points:

1.     This review article is a little difficult to read.  In order to aid for general readers to understand this article, the authors should provide an abbreviation list of the words used.

2.     Abbreviations of technical terms that are not familiar to general readers are used without explanation.  For example, they are NG2 (line 42), CC1 (line 142), PDGFR (line 181), MNI-glutamate (line 192), Olig2 (line 250), PLP1 (line 290), MAG (line 344), Cspg4 (line 362).  They should be added to the abbreviation list in comment 1 and be shortly explained if possible.

Specific points:

1.     Line 94: “D” should be “F”.

2.     Fig. 2: it may be better to mention in this legend that green “+” indicates monovalent cations.

3.     Lines 100 and 101: is English of this sentence OK?

4.     Line 104: not “72hpf” but “72 hpf”?

5.     Line 201: not “6dpf” but “6 dpf”?

6.     Line 218: “g-ratio” should be explained here but not in line 253.

7.     Line 242: “P21-70” should be “P21-P70”.

8.     Line 291: is “respectively” OK?

9.     Line 359: what is “non-ionotropic function of AMPARs”?  The phenomena just described seem to be due to AMPAR’s ionotropic fuction.

10.  Line 372: is “edited the” OK?

11.  Line 381: please explain either “Plp1” or “PLP1” (see line 290).

12.  Line 474: “(C) and” should be “and (C)”.

13.  Line 519: “AMPAR” should be “AMPARs”.

Reviewer 2 Report

The article showed an interesting topic and discuss the experimental findings of several recent studies involving in the functional role of AMPA receptors (AMPARs) in oligodendrocyte lineage cells in vivo, in mice and in zebrafish and highlighted the gap and suggested direction. However, there are a few minor issues that need to be considered:

Page 2; Line 71: What methods author apply for manuscript construction? Required to add a section as Methods or data search immediately after the introduction which will justify the construction of the manuscript in a critical way. It should include the selection criteria such as the use of what kind of database sources used (Such as Scopus, MEDLINE, PubMed, Cochrane and ScienceDirect etc.), how did the filter occur (how many years of studies, and types of study including in vitro, in vivo and clinical studies if available).

It would be great if there was a Table for the comparative analysis.

Author Response

Please, see attachment.
